# The mechanism of glycosphingolipid degradation revealed by a GALC-SapA complex structure

Chris H. Hill [1,3], Georgia M. Cook[1], Samantha J. Spratley[1,4], Stuart Fawke[1], Stephen C. Graham [2] & Janet E. Deane [1]

Sphingolipids are essential components of cellular membranes and defects in their synthesis or degradation cause severe human diseases. The efficient degradation of sphingolipids in the lysosome requires lipid-binding saposin proteins and hydrolytic enzymes. The glyco-sphingolipid galactocerebroside is the primary lipid component of the myelin sheath and is degraded by the hydrolase β-galactocerebrosidase (GALC). This enzyme requires the saposin SapA for lipid processing and defects in either of these proteins causes a severe neurode-generative disorder, Krabbe disease. Here we present the structure of a glycosphingolipid-processing complex, revealing how SapA and GALC form a heterotetramer with an open channel connecting the enzyme active site to the SapA hydrophobic cavity. This structure defines how a soluble hydrolase can cleave the polar glycosyl headgroups of these essential lipids from their hydrophobic ceramide tails. Furthermore, the molecular details of this interaction provide an illustration for how specificity of saposin binding to hydrolases is encoded.

[1] Cambridge Institute for Medical Research, Department of Pathology, University of Cambridge, Cambridge Biomedical Campus, Hills Road, Cambridge CB2 0XY, UK. [2] Department of Pathology, University of Cambridge, Tennis Court Road, Cambridge CB2 1QP, UK. [3] Present address: MRC Laboratory of Molecular Biology, Francis Crick Avenue, Cambridge Biomedical Campus, Cambridge CB2 0QH, UK. [4] Present address: Antibody Discovery and Protein Engineering, MedImmune, Cambridge CB21 6GH, UK. Correspondence and requests for materials should be addressed to J.E.D.(email: jed55@cam.ac.uk)

Sphingolipids are both essential membrane components and bioactive metabolites that regulate critical cell functions. Defects in sphingolipid metabolism underlie a range of diseases, including lysosomal storage diseases, and are implicated in a number of cancers[1–3]. Sphingolipid degradation occurs in the lysosome and depends upon two families of proteins: glycosyl hydrolases, and lipid-transfer proteins including saposins and the GM2 activator protein[4–7]. The hydrolases are water soluble while the substrates are embedded in lysosomal intraluminal vesicle membranes. The steric crowding of headgroups and lateral association of sphingolipids into clusters prevents hydrolases from accessing the scissile bonds of their target substrates. Saposins are required to facilitate sphingolipid processing by hydrolases, and extensive work provides evidence for two general models of saposin action[8–11] (Fig. 1a). The 'solubiliser' model proposes the complete extraction of lipids from the membrane, while the 'liftase' model envisages saposin proteins binding directly to compatible membranes and improving lipid accessibility by membrane distortion, destabilisation or localised remodelling. The saposin proteins are produced as a precursor prosaposin protein that, upon delivery to the lysosome, is cleaved into the four saposins: A, B, C and D[12–14]. The functions of these four saposins are distinct, as they cannot compensate for the loss of each other, and saposins appear to function specifically with their associated hydrolase[15,16]. Lysosomal storage diseases, and more specifically sphingolipidoses, are caused by mutations that inhibit degradation of sphingolipids. For example, Krabbe disease is caused by loss of galactocerebrosidase (GALC) which is responsible for the removal of the terminal galactose from the glycosphingolipid galactocerebroside (GalCer, Fig. 1b). Importantly, mutation of the GALC-associated saposin SapA can also cause Krabbe disease[17,18]. Similarly, Gaucher disease is caused by loss of glucocerebrosidase activity and also by loss of SapC, highlighting the critical role of saposins in sphingolipid processing[19–21].

Several high-resolution structures of saposins have been determined to date, revealing huge conformational variability and a propensity to form oligomeric assemblies[22–26]. Saposins can broadly be described as existing in two states: a "closed" monomeric form where the helical protein folds back on itself, burying a large hydrophobic core; or a more "open" dimeric form, possessing a hydrophobic cavity into which lipids and detergents can bind. A recent structure of SapA, solved in the presence of the detergent LDAO, revealed that this open conformation can form lipoprotein discs[22] and these discs (also referred to as picodiscs or Salipro nanoparticles) have recently been exploited to aid the determination of challenging membrane protein structures by crystallography and cryo-electron microscopy[27]. However, it has remained unclear what form of saposin–lipid complex mediates binding to their cognate hydrolase. To remedy this deficit we have solved the structure of GALC in complex with SapA, defining how saposins solubilise lipids for processing by soluble hydrolases.

## Results

**GALC and SapA form a pH-dependent heterotetrameric complex.** Murine GALC and SapA were expressed and purified from mammalian cells and *E. coli*, respectively. Pulldown assays reveal that SapA binding to GALC is pH-dependent and is optimal at low pH, equivalent to that of the endolysosomal compartments where lipid degradation occurs (Fig. 2a). This interaction depends upon the presence of detergents, suggesting that the "open" (dimeric) lipid-bound form of SapA mediates the interaction (Fig. 2b). The pH-dependency of the interaction, combined with the low pI of SapA (pH 4.5) and the higher pI of GALC (pH 6.1), suggests that the interaction could be mediated by electrostatic interactions. In support of this, the interaction is sensitive to the presence of salt, as increasing concentrations of NaCl reduce binding in these pulldown assays (Fig. 2b). Based on these insights, the GALC–SapA complex was formed at pH 5.0 in

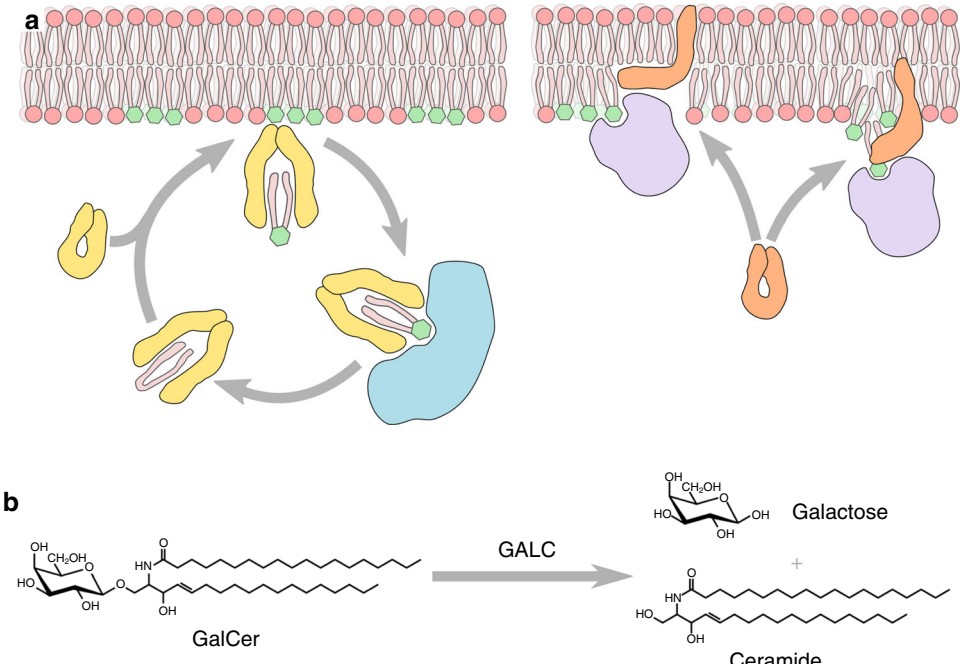

**Fig. 1** Glycosphingolipid processing by saposins and GALC. **a** Schematic diagram for the two proposed mechanisms of saposin-mediated hydrolase activation. The solubiliser model (left) proposes saposins (yellow) extract sphingolipids from the bilayer, forming a soluble saposin-lipid complex that presents the lipid to the hydrolase (blue). The liftase model (right) proposes saposins (orange) disrupt or insert into the bilayer to provide access for hydrolases (purple) to the lipid substrates at the bilayer surface. Glycosyl headgroups are illustrated as green hexagons. **b** The cleavage of galactocerebroside (GalCer) by GALC produces galactose and ceramide

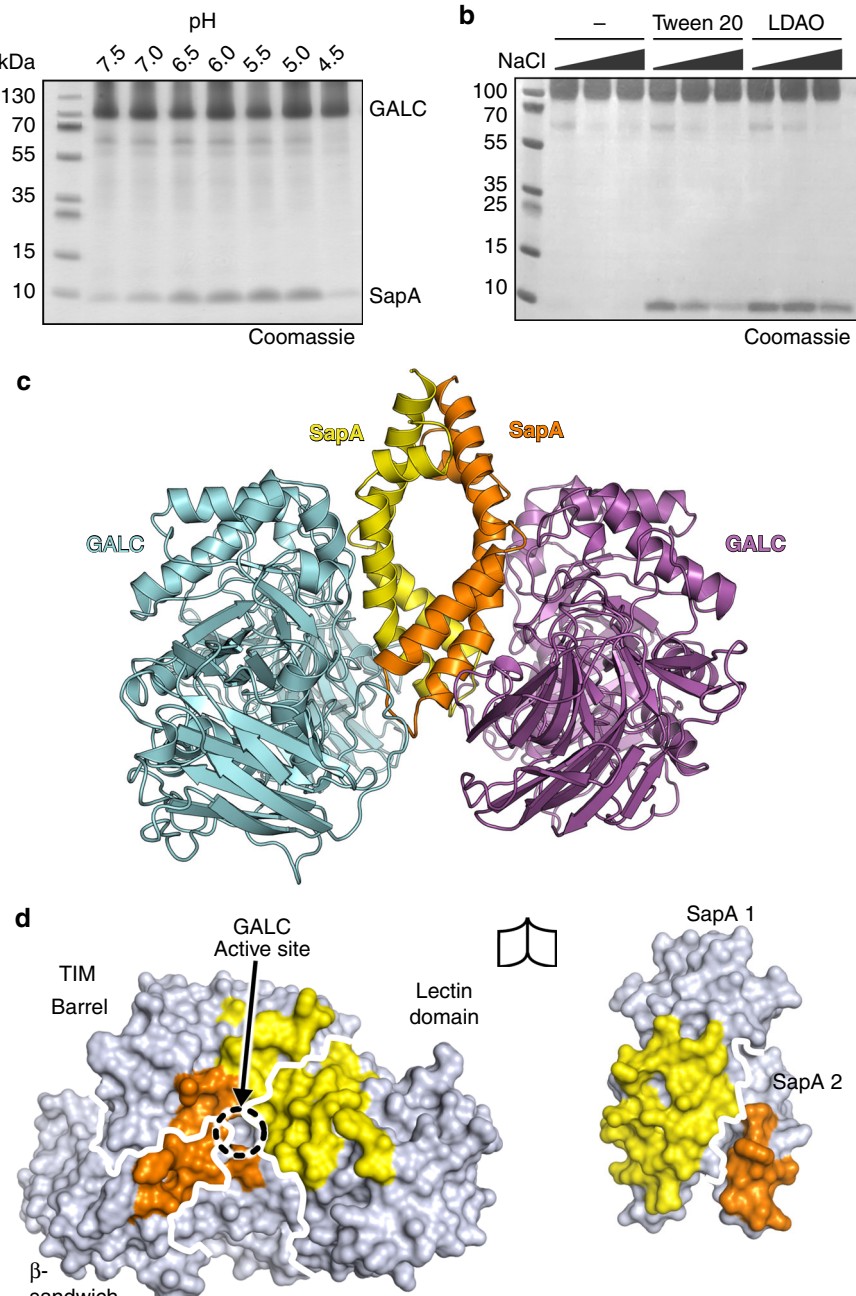

**Fig. 2** GALC–SapA complex formation and structure. **a** Coomassie-stained SDS–PAGE of pulldown assays with immobilised GALC across the pH range 7.5 to pH 4.5 in the presence of detergent. The intensity of SapA binding to GALC is 3.6 times greater at pH 5.5, equivalent to the late-endosomal/lysosomal compartment ($n = 3$). Glycosylation of GALC results in a smeared band on gradient SDS-PAGE gels. **b** Pulldown assays at pH 5.4 identify that the interaction requires the presence of detergents, such as Tween 20 or LDAO, and is reduced with increasing NaCl concentration (20, 150 and 300 mM NaCl). **c** The crystal structure of the GALC–SapA complex reveals a pseudo-symmetric 2:2 heterotetramer of SapA (yellow and orange) with GALC (cyan and magenta). **d** The interacting surfaces of GALC (left) and the SapA dimer (right), rotated by 90° with respect to **c** to highlight the respective interaction surfaces. On the GALC surface the interacting regions contacted by SapA chain 1 (yellow) and SapA chain 2 (orange) are highlighted, the different domains of GALC are outlined (white lines) and the active site is identified. On the SapA surface the residues that interact with GALC are highlighted for each chain and the different chains of SapA are outlined (white line)

the presence of the detergent LDAO and the X-ray crystal structure of this complex solved and refined to 3.6 Å resolution (Fig. 2c and Table 1). The structure was solved by molecular replacement using the high-resolution structure of GALC ($d_{min} = 1.97$ Å, PDB 4CCE[28]) as a search model. After placement of two GALC molecules, strong $F_O$–$F_C$ density was evident and could be readily interpreted as two molecules of SapA (Supplementary Fig. 1a).

The structure reveals that the complex forms a 2:2 heterotetramer, composed of a central SapA dimer with two molecules of GALC binding its surface. The SapA–GALC heterodimers are related to each other by a pseudo two-fold non-crystallographic symmetry axis. Using a high-resolution GALC structure, and thus high-quality phase information, to solve the complex structure yielded excellent maps: the SapA molecules could be unambiguously positioned due to density for their intramolecular

**Table 1 Data collection and refinement statistics**

| Data collection | |
| --- | --- |
| Space group | $P\,6_2\,2\,2$ |
| Cell dimensions | |
| $a,b,c$ (Å) | 187.2, 187.2, 360.2 |
| Resolution (Å) | 180.12–3.60 (3.70–3.60) |
| $R_{merge}$ | 0.386 (2.946) |
| $R_{pim}$ | 0.078 (0.607) |
| $CC_{1/2}$ | 0.945 (0.677) |
| $I/\sigma I$ | 9.3 (1.5) |
| Wilson B (Å$^2$)$^a$ | 94.5 |
| Completeness (%) | 100 (100) |
| Redundancy | 25.6 (25.2) |
| **Refinement** | |
| Resolution (Å) | 162.11–3.60 |
| No. reflections in working set | 43,825 |
| No. reflections in test set | 2138 |
| $R_{work}/R_{free}$ | 0.217/0.231 |
| No. atoms | |
| Protein | 11,522 |
| Other$^b$ | 220 |
| B-factors (Å$^2$) | |
| Protein | 100.9 |
| Other$^b$ | 141.2 |
| Ramachandran | |
| Favoured (%) | 95.8 |
| Outliers (%) | 0.5 |
| r.m.s. deviations | |
| Bond lengths (Å) | 0.008 |
| Bond angles (°) | 0.94 |

Values in parentheses are for highest-resolution shell
$^a$Calculated using phenix.model_vs_data[59]
$^b$Includes carbohydrate and ions

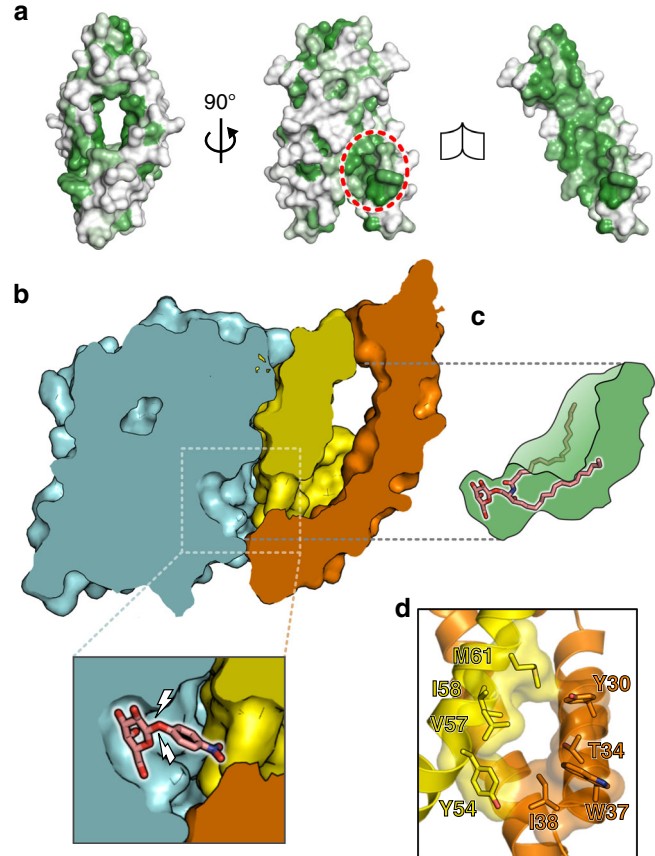

**Fig. 3** The GALC–SapA structure reveals a hydrophobic channel buried in the core of the complex. **a** Residue hydrophobicity (green) is mapped onto the surface of the SapA dimer. The left panel is in the same orientation as shown in Fig. 2c, while the central panel is rotated by 90° to be in the same orientation as Fig. 2d. The region that binds over the GALC active site is circled (red dashed line). The right panel maintains the orientation of the central panel, with one SapA chain removed to reveal the highly hydrophobic inner surface of the SapA dimer. **b** Cross-section through the GALC–SapA structure reveals an open channel stretching from the GALC active site into the SapA hydrophobic cavity. The GALC surface and cut-through is shown in cyan and the SapA dimer surface is shown in yellow and orange. For clarity, the second GALC monomer is not shown. An expanded view of the active site (bottom left panel) illustrates how substrate binds in this pocket based on the previously determined substrate-bound GALC structure (PDB ID 4CCC,[28]) and identifies the scissile bond (white bolts). **c** Illustration of the hydrophobic cavity (green) showing that the lipidated substrate GalCer (pink sticks) can fit into this cavity, bridging the channel from the GALC active site to hydrophobic SapA core. **d** Residues that form the opening at the SapA dimer surface opposite the GALC active site are shown as sticks with a semi-transparent surface. The displayed region of the SapA dimer is the same as that circled in panel **a**

disulphide bonds, and small GALC conformational changes in the vicinity of the bound SapA molecules were readily visible (Supplementary Fig. 1b–d). By peeling apart the complex structure the details of the surfaces involved in the interaction between GALC and SapA are revealed (Fig. 2d). Importantly, the active site of GALC is buried in the centre of the interaction interface. Not only does the SapA dimer sit over the active site, it makes contact with all three domains of GALC: the TIM barrel, β-sandwich and lectin domains[29]. This extensive interface buries a total surface area of 1366 Å$^2$, of which 70% is contributed by one SapA chain and 30% by the other (yellow and orange, respectively, in Fig. 2d). At the optimal pH of the interaction the GALC and SapA surfaces demonstrate extensive charge complementarity, explaining the observed pH dependency of the interaction (Supplementary Fig. 2). The GALC–SapA interaction interface is not in the vicinity of any crystal contacts and thus is not affected by surrounding molecules in the crystal (Supplementary Fig. 1e).

**The GALC–SapA complex possesses a hydrophobic channel.** Analysis of the SapA dimer in the complex structure suggests the presence of disordered detergent within the core. Hydrophobic residues line the SapA core, consistent with this surface being exposed to non-polar solvent (Fig. 3a, the right panel reveals the internal surface of the dimer). Further, the average electron density in the cavity is lower than the surrounding solvent, consistent with the presence of detergent (Supplementary Fig. 3a). Analysis of the SapA dimer surface reveals a clear hydrophobic patch, within the interaction interface, at the point where the two SapA chains come together to contact the GALC surface (Fig. 3a, circled in the central panel, and Fig. 2d). A cross-section through the structure at this position shows that this hydrophobic patch

encircles an opening in the SapA dimer surface that lies directly opposite the GALC active site (Fig. 3b). This reveals a continuous open channel that stretches from the GALC active site through to the hydrophobic cavity buried in the SapA dimer. The hydrophobic acyl chains of glycosphingolipids such as GalCer can thus be shielded from the polar solvent by the SapA dimer, while the hydrophilic glycosyl head groups are presented to the GALC active site. Although GALC–SapA crystals could be obtained in the presence of lipids and substrate-mimics, we were not able to collect data of sufficient quality to yield a structure of the substrate-bound complex. However, our previous work

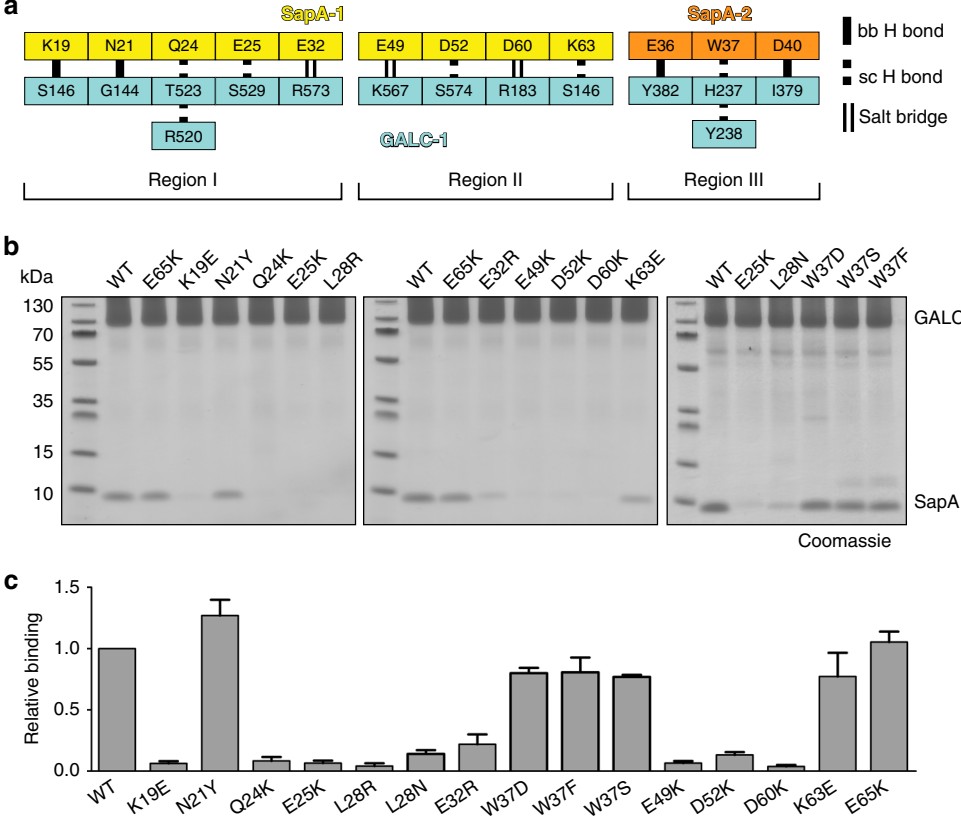

**Fig. 4** Critical residues at the GALC–SapA interface. **a** Schematic diagram illustrating the backbone (bb) hydrogen bonds, sidechain (sc) hydrogen bonds and salt bridges observed between GALC (cyan) and specific residues of SapA chains 1 (yellow) or 2 (orange). Note that SapA residues Q24 and W37 interact with multiple GALC residues. **b** Coomassie-stained SDS–PAGE of pulldown assays with immobilised GALC performed in the presence of Tween-20 identifies critical SapA mutations at the interface that abolish GALC binding in solution. **c** Quantification of SapA binding in pulldown assays ($n = 3$, error bars represent SEM)

identifying how a bona fide synthetic substrate binds the GALC active site (Fig. 3b inset) was used to guide the placement of a lipidated substrate in this channel[28]. The position of the galactose headgroup of GalCer and the orientation of the scissile bond can therefore be reliably defined and is compatible with the acyl chains being oriented into the hydrophobic channel. Previous work has shown that, upon binding of substrate, GALC residues E182 and R380 undergo conformational changes to allow positioning of the remainder of the substrate leaving group[28]. Although this plasticity near the active site and the conformational flexibility of the GalCer acyl chains hindered accurate modelling of the conformation of an entire GalCer molecule, the dimensions of the channel and hydrophobic cavity could accommodate the lipidated substrate (Fig. 3c). Based on the SapA lipoprotein disc structure containing ordered LDAO molecules, the volume of the SapA dimer cavity in our structure is sufficient to contain between 10 and 14 molecules of LDAO (Supplementary Fig. 3b) and therefore would be capable of binding at least 2 molecules of GalCer, meaning both GALC molecules in the complex could function simultaneously. The opening in the SapA dimer surface through which the GalCer acyl chains must pass is surrounded by both hydrophobic and polar residues from both SapA chains (Y30, T34, W37 and I38 from one SapA, and Y54, V57, I58 and M61 from the second SapA, Fig. 3d). The orientation of W37 and I38 differs from that of the lipoprotein disc structure and is influenced by interactions with the GALC surface. As was observed for substrate-bound GALC, it is likely that additional residues in this region may alter their conformation in order to accommodate lipidated substrate. In particular, residue

Y54, which is highly conserved across the saposin family, forms the limiting edge of the channel nearest the active site and thus may play a critical role in substrate positioning. The presence of this continuous channel from the active site to the hydrophobic cavity, with sufficient space to accommodate glycosylated sphingolipid substrates, supports that this arrangement of GALC and SapA represents the functional complex that facilitates glycosphingolipid catabolism in vivo.

**Specific mutations of SapA inhibit binding to GALC.** Analysis of the molecular details at the interface reveals that several sidechains of residues along the length of one SapA chain form critical hydrogen bonds, electrostatic and hydrophobic interactions with residues in GALC (Fig. 4a and Supplementary Fig. 4). These interactions span two stretches of this SapA chain: residues 19–32 and 49–63. The interactions contributed by the second SapA chain encompass residues 36–40 and involve several backbone hydrogen bonds, as well as sidechain interactions with W37. To confirm that the interface observed in our crystal structure mediates the interaction in solution, a series of point mutations were made in SapA and tested for their ability to bind GALC (Fig. 4b, c). Several of the SapA residues that lie at the interface with GALC are acidic: E25, E32, E49, D52 or D60. Single-charge inversions at these sites each abolish or significantly reduce binding to GALC, highlighting the electrostatic nature of the interface. To ensure that alteration of surface charge alone is not sufficient to abolish binding we mutated residue E65, which does not lie in the interface, to lysine and confirmed it does not

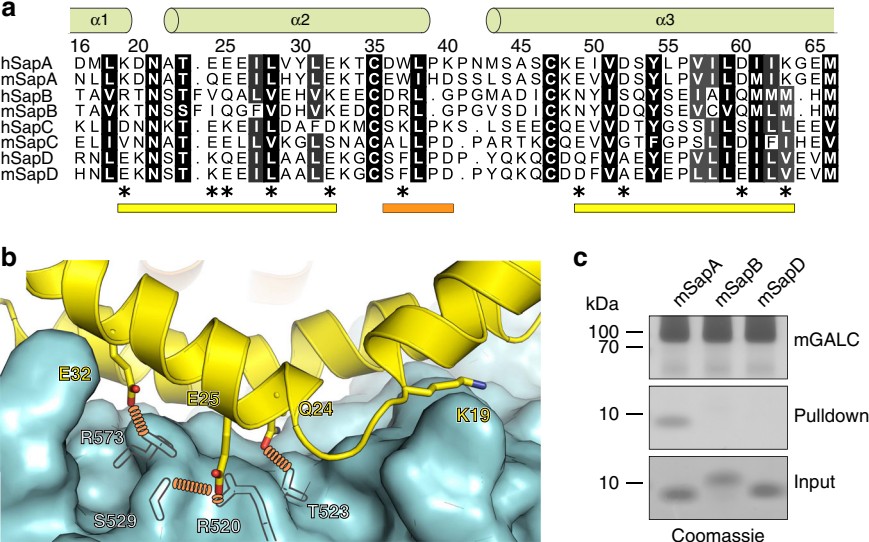

**Fig. 5** Saposin specificity. **a** Sequence alignment of human and murine saposins A–D coloured by conservation. Important residues at the GALC–SapA interface are highlighted (\*). Coloured bars below the sequence identify the regions that interact with GALC: two regions from SapA chain 1 (yellow) and the one from SapA chain 2 (orange) as detailed in Fig. 4a. The position of α-helices in SapA are highlighted (green cylinders). **b** Critical residues that may determine binding specificity are shown (sticks) on SapA (yellow) at the interface with GALC (cyan surface, key residues depicted as transparent sticks). Hydrogen bonds between GALC and SapA residues are highlighted (orange dotted lines). **c** Coomassie-stained SDS-PAGE pulldowns with immobilised murine GALC (mGALC) illustrating specific binding of murine SapA (mSapA), but not mSapB or mSapD

interfere with SapA binding to GALC. In addition to electrostatic interactions, hydrophobic interactions at the interface are also critical for binding. The sidechain of L28 interacts with a hydrophobic surface of GALC (Supplementary Fig. 4c) and mutation of this residue to either arginine or asparagine significantly reduces SapA binding to GALC. Residue N21 lies at the interaction interface and forms a backbone hydrogen bond with G144 (Supplementary Fig. 4c). In the context of the cell this highly conserved residue is post-translationally glycosylated and the position of this sidechain at the surface of GALC is compatible with this modification. In an attempt to mimic this bulky modification, which is capable of both hydrophobic and hydrogen-bond interactions, N21 was mutated to tyrosine. This mutation did not interfere with SapA binding to GALC, consistent with the binding of glycosylated SapA to GALC in vivo. Residue 24 is a glutamine that forms hydrogen bonds with residues T523 and R520 of GALC (Fig. 4a and Supplementary Fig. 4c). In human SapA the corresponding residue is a glutamate, which could form a salt bridge with R520. Mutation of Q24 to lysine blocks SapA binding to GALC, supporting its role in stabilisation of the interaction. The backbone of residue K19 forms a hydrogen bond with GALC residue S146, while the acyl region of the K19 sidechain forms hydrophobic interactions with the GALC surface. It is likely that the disruption of this hydrophobic interaction, rather than the charge inversion by substitution of K19 for glutamate, abolishes the GALC–SapA interaction. The only sidechain from the second SapA molecule that interacts with the GALC surface is W37. This large sidechain contributes both hydrophobic and hydrogen bond interactions at the interface and several mutations of this residue were tested for their ability to bind GALC. All three mutations W37S, W37D and W37F maintained close to wild-type levels of binding suggesting that, in the context of this pulldown, the contribution of W37 is less critical than that of other interface residues (Fig. 4). However, W37 forms part of the opening through which the acyl chains of the substrate must pass (Fig. 3d). This suggests that, although it may not be essential for the interaction with GALC, W37 is likely to play a critical role in substrate positioning. Taken together, our

mutational analysis highlights the crucial importance of SapA residues K19, Q24, E25, L28, E32, E49, D52 and D60 for complex formation.

**GALC binds specifically to SapA**. There is high structural similarity between the different saposin proteins[30], suggesting that the specific binding to their cognate enzyme is sequence-mediated. The identification here of specific residues essential for SapA binding to GALC provides a framework for understanding this specificity. Sequence alignments of saposins A-D reveal the limited conservation between saposins and identifies several charge inversions equivalent to those used in this study, particularly in the region spanning residues 19–32 (Fig. 5a and b). Based on our binding data, single point mutations in SapA are sufficient to inhibit binding to GALC and thus the limited conservation with saposins B, C and D suggests they would be unable to bind GALC. While we were unable to purify murine SapC, we observed that murine GALC (mGALC) binds specifically to SapA but not to murine SapB (mSapB) or mSapD (Fig. 5c). We were able to purify small quantities of human GALC (hGALC) and human SapC (hSapC), allowing us to test their binding in pulldown assays. While we could detect GALC binding to SapA, there was no detectable direct binding to SapC (Supplementary Fig. 5). Thus, despite the high structural homology of the saposin proteins it is their distinct sequences across the three regions identified here as involved in binding to GALC that determine the binding specificity for their cognate hydrolases.

**Discussion**
Here we reveal molecular details of how glycosphingolipids can be bound by SapA and presented to the active site of GALC for catabolic processing. The interaction of SapA with GALC is pH dependent and optimal at pH 5–6, consistent with its activity in the late endosomal/lysosomal compartments. This pH dependence can be understood by enhanced charge complementarity between GALC and SapA at pH 5.4 (Supplementary Fig. 2). The GALC–SapA interaction also requires the presence of detergents, a proxy for the biological lipid substrates. Structures of SapA in

the absence of detergents remain closed[24,31] whereas in the presence of detergents SapA adopts an open conformation (this study and Popovic et al.[22]), suggesting that detergent mediates the formation of open SapA dimers essential for the interaction. The crystal structure of GALC–SapA determined here reveals the complex to be a heterotetramer with an extensive interaction interface that involves residues from all domains of GALC and both chains of the SapA dimer. There is an open channel running from the GALC active site through to the hydrophobic cavity of the SapA dimer, suggesting a clear mechanism for glycosphingolipid binding and processing. The combination of structure analysis, pulldown assays and sequence comparison of the saposin family members identifies residues 19, 24, 25, 28, 32, 49, 52 and 60 of SapA as essential for binding to GALC. Despite the structural homology of the saposin proteins it is clear that the distinct charge and shape complementarity of the SapA surface is critical for specific binding to GALC.

The specific importance of SapA for mediating GALC activity in vivo is highlighted by two key observations: clinical data has identified that a mutation in SapA causes Krabbe disease despite normal GALC activity[17], and a transgenic mouse with defective SapA develops a phenotype resembling late-onset Krabbe disease[18]. In support of this specificity, mutations that result in defective SapC cause Gaucher disease, not Krabbe disease[19–21]. However, in vitro studies of saposin-mediated degradation of galactosphingolipids have shown that both SapA and SapC can stimulate GALC activity[32,33]. Our pulldown studies using recombinant GALC and saposins (Fig. 5c and Supplementary Fig. 5) suggest that, unlike SapA, SapC does not directly bind GALC. Taken together, this suggests that although SapC is not necessary for GALC activity in vivo it may indirectly enhance GALC activity by, for example, aiding lipid accessibility at the bilayer surface.

The GALC–SapA complex structure reveals new details of how saposins bind to lysosomal hydrolases. However, it is not the first SapA dimer structure and comparison with the previous structure of a human SapA lipoprotein disc reveals significant differences in dimer assembly[22]. Interestingly, the conformation of the SapA monomers in these structures are essentially identical (RMSD 0.86 Å² over 80 Cα atoms, with 79.7% sequence identity), adopting the same hinge angle upon dimer formation (Supplementary Fig. 6a). Despite this similarity, the relative orientations of the monomers within the dimer differs between these structures (Supplementary Fig. 6b). In our structure the monomers are head-to-head, with all termini together at one end, while in the published lipoprotein disc structure the monomers adopt a head-to-tail orientation. However, molecular dynamics studies of the previous lipoprotein disc structure and subsequent work monitoring the dynamics of these particles shows there is significant plasticity in the arrangement of the SapA molecules as their relative orientations are only determined via binding to LDAO (Supplementary Fig. 6c)[22,34,]. In complex with GALC, the SapA dimer possesses direct protein-protein contacts between saposin monomers, including extensive hydrophobic interactions where the termini come together and additional contacts near the interface with GALC (Supplementary Fig. 6d). Comparison of the different SapA dimer structures shows that the GALC-interacting surface identified in our structure is surface exposed in the lipoprotein disc structure. Docking of GALC onto the surface of the lipoprotein disc structure reveals this SapA dimer to be compatible with GALC binding (Supplementary Fig. 7a). In this alternative arrangement the opening on the surface of the SapA dimer is less hydrophobic and, due to the loss of interactions with the second SapA chain, the substrate channel is more open and exposed to surrounding solvent (Supplementary Fig. 7b, c). Although there are no steric clashes that would prevent this

complex forming, the tighter SapA dimer and enclosed hydrophobic channel identified in our complex provides a more protected environment from the hydrophilic solvent. We therefore propose that our structure, possessing a large buried surface area, an enclosed hydrophobic channel and an obvious mechanism for substrate presentation to GALC, represents the *bona fide* interaction that stimulates SapA-mediated lipid catabolism in the lysosome.

As both open and closed forms of SapA have been described we considered whether the closed, monomeric conformation could be compatible with binding to GALC. Closed SapA can be docked onto the open form of SapA via either the terminal helices or the central two helices (Supplementary Fig. 8a). Only this second alignment yields an interface that maintains any of the critical interactions identified in our pulldowns (Supplementary Fig. 8b). This arrangement would block binding of a second SapA and GALC, resulting in a 1:1 complex possessing a buried surface area of 669 Å² (compared with 1366 Å² for the tetramer) that is not predicted by ePISA to form a stable complex (CSS = 0)[35]. Thus, although there is no steric hindrance to binding of closed SapA to GALC it is likely to be an extremely weak interaction, below the detection limit of our assays, if it exists at all.

Previous work has identified the pathogenic mechanisms underlying specific disease-causing mutations of GALC, including protein misfolding and active site defects[28,36]. Two mutations responsible for causing Krabbe disease, E215K and P302R, have been shown to traffic correctly in cells, identifying that they are correctly folded, and retain enzymatic activity[36–39]. These two mutations lie on the GALC surface adjacent to the SapA interaction site (Supplementary Fig. 9). Although these residues do not form direct interactions with SapA residues, their proximity to the binding site and their mutation to bulky, charged sidechains suggests these mutations may alter the surface shape and surface charge of GALC, interfering with binding to SapA. Thus, an inability to bind their cognate saposin may represent a new pathologic mechanism for sphingolipidoses.

Our structure reveals how saposins and hydrolytic enzymes can interact and can be compared with the recent structures of acid sphingomyelinase (ASM), which possesses an intramolecular saposin-like domain[40–42]. Although the enzymatic portion of ASM is not structurally similar to GALC, the saposin-like region adopts a similar conformation to SapA in the GALC–SapA complex, allowing a reliable overlay to be made based on this portion (Supplementary Fig. 10a). This comparison reveals that the catalytic domains are in similar positions relative to the saposin (or saposin-like domain) and that the enzyme active sites are adjacent to the saposin, although not directly overlaid (Supplementary Fig. 10b). However, there are significant differences between these structures and this may reflect the different mechanisms by which the saposin domains facilitate lipid processing. Specifically, SapA is thought to function as a 'solubiliser', an assertion strongly supported by both the GALC–SapA structure presented here and the SapA lipoprotein disc structure[22]. ASM, on the other hand, is likely to function via a 'liftase' mechanism. Specifically, ASM has been shown to bind tightly to negatively charged membranes and the saposin domain is likely to play a critical role in orienting the active site towards the membrane[41,43]. Dimerisation of the saposin-like domain would not be required in this 'liftase' model of ASM activity, and indeed we observe that steric hindrance would prevent formation of a SapA-like dimer by the ASM saposin-like domain. We therefore conclude that the mechanisms used by saposins to facilitate lipid presentation will influence the stoichiometry and topology of their association with their cognate hydrolases.

## Methods

**Protein expression and purification.** His[6]-tagged murine GALC (mGALC) was expressed using our stably transfected HEK 293 T cell line and purified from conditioned medium using nickel-affinity chromatography (Ni-NTA agarose resin) in phosphate-buffered saline pH 7.4[28,31]. Untagged murine SapA (mSapA) was expressed in *Escherichia coli* Origami (DE3) cells and purified as follows. Cleared lysate was heat-treated (100 °C, 30 min), precipitated proteins were cleared by centrifugation (40,000 g, 40 min) and supernatant containing mSapA was dialysed overnight in the presence of 20 μg mL⁻¹ DNAse against anion exchange buffer (50 mM Tris pH 7.4, 25 mM NaCl). mSapA was further purified by anion exchange chromatography (HiTrap QSepharose column) followed by size-exclusion chromatography (HiLoad 16/600 Superdex 75 column) in 50 mM Tris pH 7.4, 150 mM NaCl[28,31]. mSapA mutants were made by site-directed mutagenesis and verified by sequencing (Supplementary Table 1). Mutant proteins were expressed and purified exactly as for wild-type. Purified mGALC was concentrated to 5–15 mg ml⁻¹ and stored in phosphate-buffered saline pH 7.4 at 4 °C. Purified mSapA was concentrated to 8.0 mg ml⁻¹ and stored in 50 mM Tris pH 7.4, 150 mM NaCl at 4 °C. mSapB, mSapD and hSapC cDNA was synthesised (GeneArt) and cloned into the pET15b vector using NcoI and XhoI restriction-endonuclease sites. Untagged mSapB, mSapD and hSapC were expressed and purified as for mSapA[31]. Human GALC (hGALC) was expressed and purified as described above with minor modifications as follows[28,29]. The cDNA encoding hGALC was obtained from the IMAGE clone library and cloned into pSecTag2B with an N-terminal His tag and Factor Xa site and confirmed by sequencing. This construct was transfected into HEK 293 T cells and a cell line generated by clonal selection that stably expressed hGALC. Protein was purified by nickel-affinity chromatography, as described above for mGALC[29].

**Pulldowns.** mGALC was mixed with magnetic Ni-NTA agarose beads under saturating conditions (30 μl beads, 75 μg GALC per experiment) and incubated with mixing (90 min, 4 °C). Loaded beads were transferred to a flat bottomed 96-well plate and washed twice in 200 μl pulldown buffer containing 20 mM NaCl, 0.05% Tween-20 or LDAO and appropriate buffer. For the pH screen 100 mM phosphate buffer encompassing the pH range 4.5–7.5 was used while for subsequent pulldowns 100 mM sodium acetate pH 5.4 was used to more closely match that used in crystallisation experiments. Concentrated SapA was pre-incubated with 0.1% Tween-20 or LDAO for 1 hour and 160 μg of SapA was added to GALC-loaded beads in 200 μl pulldown buffer and incubated with shaking (2 hr, 4 °C). Beads were then washed for 60 s four times with 200 μl pulldown buffer. Proteins were eluted with 40 μl 500 mM imidazole, PBS pH 7.4, 0.05% Tween-20 and analysed by 4–12% gradient Bis-Tris PAGE. Following staining with Coomassie, gels were scanned on an Odyssey imaging system and band intensity determined using Image Studio Lite (LI-COR Biosciences). Uncropped gel images from pulldown experiments are provided in Supplementary Fig. 11.

**Protein complex crystallisation.** Purified mSapA was pre-incubated with LDAO in a final mix of 180 μM mSapA with 50 mM sodium acetate pH 5.0, 700 mM NaCl and 0.1% LDAO. mGALC was concentrated to 90 μM in PBS. Equal volumes of the mSapA–LDAO complex and purified mGALC were combined to give a final SapA: GALC molar ratio of 2:1 and incubated at room temperature for 15 min. This complex was then diluted 1:2 with water for crystallisation trials. Crystallisation experiments were performed in 48-well sitting drops (800 nL of complex as prepared above plus 800 nL of precipitant) equilibrated at 20 °C against 200 μL reservoirs of precipitant. Diffraction quality crystals grew against a reservoir of 75 mM sodium citrate pH 5.6 and 11% (w/v) PEG 3350. Crystals were cryoprotected in reservoir solution supplemented with 20% (v/v) glycerol and flash-cooled by plunging into liquid nitrogen.

**X-ray data collection and structure determination.** Diffraction data were recorded at Diamond Light Source beamline I04-1 on a Pilatus 6 M detector (Dectris). Diffraction data were collected at 100 K. Data collection statistics are provided in Table 1. Diffraction data were indexed and integrated using DIALS and scaled and merged using AIMLESS the xia2 automated data processing pipeline[44]. Resolution cut-off was decided by $CC_{1/2}$ value of > 0.5 and $I/\sigma I$ of 1.5 in the outer resolution shell[45,46]. The structure was solved by molecular replacement using Phaser with mouse GALC (PDB ID 4CCE[28]) as a search model. Placement of a model of mouse SapA, based on the structure of the human SapA in a lipoprotein disc (PDB ID 4DDJ[47]), and further manual model building was performed using COOT[48]. The structure was refined using autobuster[49], over-fitting of the data being minimised by the use of local structure similarity restraints[50] to the high-resolution structure of mouse GALC (PDB ID 4CCE[28]) and between the NCS-related GALC and SapA monomers. Model geometry was evaluated with MolProbity throughout the refinement process[51]. Final refinement statistics are presented in Table 1. Structural figures were rendered using PyMOL (Schrödinger LLC).

**Structure analysis.** The analysis of buried surface area and interface interactions in the GALC–SapA structure was carried out using the ePISA service at the European Bioinformatics Institute, EBI[35]. Hydrogen bonding representations were created using LIGPLOT + implementing DIMPLOT to generate schematic diagrams of protein–protein interactions[52]. Structure-based alignments were carried out using SSM superposition implemented within COOT[53]. Multiple sequence alignments were carried out using the MUSCLE service at the EBI[54]. The figure of saposin sequence alignment and conservation was produced using ALINE[55]. For the electrostatic potential calculations partial charges were assigned using PDB2PQR[56], which uses PROPKA[57] to determine protein pKa values. Electrostatic surfaces were calculated using APBS[58].

**Data availability.** The atomic coordinates and structure factors for the GALC–SapA complex have been deposited in the Protein Data Bank under accession code 5NXB. Other data are available from the corresponding author upon reasonable request.

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

## Acknowledgements

We acknowledge Diamond Light Source for time on beamlines I04 and I04-1 under proposal MX11235. Remote access was supported in part by the EU FP7 infrastructure grant BIOSTRUCT-X (Contract No. 283570). The Cambridge Institute for Medical Research is supported by Wellcome Trust Strategic Award 100140. J.E.D. is supported by a Royal Society University Research Fellowship (UF100371). This work is supported by MRC grant MR/N020626/1 to J.E.D. S.C.G. is supported by a Sir Henry Dale Fellowship co-funded by the Royal Society and Wellcome Trust (098406/Z/12/Z).

## Author contributions

C.H.H., S.C.G. and J.E.D. designed the experiments, performed experiments, analysed the data and wrote the paper; G.C., S.J.S. and S.F. performed additional experiments and helped write the paper. All authors participated in the discussion and interpretation of the results.

## Additional information

**Competing interests:** The authors declare no competing financial interests.

