## [Peer Review File · Nature Communications]

Reviewers' comments:

Reviewer #1 (Remarks to the Author):

Lysosomal degradation of sphingolipids by hydrolases is assisted by saposins, which facilitate access to lipid substrates through various means. Hill and co-workers report the first crystal structure of a lysosomal hydrolase (GALC) in complex with a saposin protein (SapA) at 3.6 Å resolution. The heterotetrameric structure (2 GALC:2SapA) reveals an electrostatic interface responsible for saposin specificity and a connecting channel between the GALC active site and the hydrophobic cavity of the saposin dimer, providing a model of lipid presentation to the enzyme. The GALC-SapA interface is tested by mutagenesis and in vitro pulldown assays.

Overall the results are interesting, the paper is well written, concise and easy to read. It was particularly satisfying to see Fig. 2B, C which illustrates very nicely the likely manner of lipid delivery to the GALC active site. Although this mechanism appears quite convincing based on the structure alone, it still needs to be tested, which is the major weakness of the paper - the model would be much stronger if it could be tested through mutagenesis of the proposed lipid binding cavity combined with, at the very least, an in vitro lipid hydrolysis assay. Is there any evidence in vivo (or even in vitro) that the two proteins form a heterotetrameric complex and is this complex catalytically active to begin with? Does SapA binding regulate the activity of GALC?

Beyond this major point, the following are a list of minor points that should be addressed:

- 1) Can the authors explain based on the structure, why the closed form of SapA cannot interact with GALC?
- 2) There is no discussion on what could occur after lipid hydrolysis is complete. What regulates docking and undocking of SapA from GALC?
- 3) Does lipid access to the active site occur symmetrically on both protomers? In other words, does the SapA dimer have enough room for two lipid molecules? It is not clear from Supp. Fig. S2 if this is the case. Or can only one catalytic event occur at a time?
- 4) The R-factors (particularly R_{free}) are unusually low for 3.6 Å resolution. I realize the refinement was restrained to the high resolution structure of GALC, but was this also done for SapA? How was the R_{free} test set chosen? Also in Table 1: the Wilson B-factor should be added to the data collection statistics.
- 5) Title: "The mechanism of sphingolipid processing revealed by a GALC-SapA complex structure" - is a bit overreaching. Various sphingomyelinases and ceramidases employ different mechanisms. The authors should consider something more specific such as: "The mechanism of glycosphingolipid degradation..."
- 6) Introduction: "lipid-transfer proteins including saposins and GM2 activator proteins" - there is only one GM2 activator protein.
- 7) Introduction: "polycistronic prosaposin" - polycistronic refers to multiple open reading frames on one mRNA. It should be changed to "precursor prosaposin".
- 8) Results: "In addition to electrostatic interactions, hydrophobic interactions at the interface are also critical for binding. The sidechain of L28 interacts with a hydrophobic surface of GALC (Figure S3C) and mutation of this residue to arginine abolishes SapA binding to GALC."

This mutation causes likely results in unavoidable steric clashes with GALC, so the importance of the 'hydrophobic' nature of the interaction involving L28 cannot be assessed by it.

9) Results: "Residue N21 lies at the interaction interface and forms a backbone hydrogen bond with G144 and a number of hydrophobic interactions with the GALC surface. However, in the context of the cell this highly-conserved residue is post-translationally glycosylated. N-linked glycans often interact with hydrophobic patches on the surface of proteins (29) and mutation of N21 to the bulky hydrophobic sidechain tyrosine moderately increased SapA binding to GALC, suggesting that the glycosylation of N21 contributes to SapA-GALC binding in vivo."

N21 is facing GALC with its polar side chain, and that area of GALC is mainly lined with backbone atoms. Water molecules are present there, in the high resolution structure of GALC (PDB 4CCD). Therefore, hydrophobic interactions are probably not at play here; is there evidence that glycosylation of N21 actually contribute to complex formation?

10) Discussion: "The interaction of SapA with GALC is pH dependent" - based on the structure of the complex, is there an explanation for this?

11) Discussion: "Specifically, SapA is thought to function as a 'solubiliser', an assertion strongly supported by the GALC - SapA structure presented here." - References should be added that support this model (such as #21, where soluble SapA lipoprotein particles are described).

12) Figure 2: the channel connecting the GALC active site to the hydrophobic cavity in the saposin dimer seems to be narrowed down by the side chain of saposin Y54. Is this the case? Does the substrate still fit? Could rotation of this side chain facilitate substrate entry?

13) Figure 3A: the legend should specify that some SapA residues interact with multiple GALC residues; otherwise parts of the diagram could be interpreted as interactions within GALC residues.

14) Fig. 3B: has W37 also been mutated, to assess the importance of the secondary interaction surface?

Reviewer #2 (Remarks to the Author):

Hill et al present the crystal structure of a complex between galactocerebrosidase (GALC) and its activator protein, SapA. This is an important "landmark" structure that explains a lot of preceding biology/biochemistry. To their credit, the authors did not place a hold on the PDB coordinates (5NXB), and this made it easier to review this work. The MS does a good job of describing this remarkable complex.

Major comments:

1) Several SapA residues are tested for the importance with their pulldown assay (Fig. 3). The results are convincing (unfortunately, it would be more difficult to carry out a similar test of the GALC surface-exposed residues since GALC is produced in mammalian cells - SapA is expressed in E. coli). The GALC-SapA interface has many electrostatic salt bridges between complementary residues and this is extensively documented and discussed. However, the interface is more complex than charge complementarity and contacts that appear to be important are glossed over. In particular, the "region 2" surface of SapA (SapA-2/GALC) appears to be highly important but is largely dismissed because it involved mostly backbone residues. This is an oversimplification - this surface of SapA has excellent

shape complementarity to the GALC surface and contributes a large percentage of the buried surface. I disagree with the statement: "... residues 36-40 primarily participates in backbone interactions so does not determine specificity" (lines 152-153). The side chains in this region have important roles in positing the backbone residues for complex formation.

Overall, I would suggest a more balanced interpretation of all three SapA regions of the binding surface (see also points 2 and 3 below).

2) Of particular note, the side chain of SapA W37 in region 2 makes extensive contacts with GALC (Figs 3, S3). Can this residue be tested with a mutant? The mutants tested in Fig. 3 cover one chain only.

3) "Charge inversion" between the saposins is overemphasized throughout - this implies negative selection of the other saposins to GALC. Given the low sequence identity between the four saposins, it is just as likely that the differences are purely due to neural drift or positive selection to the partner enzymes. The argument in lines 153-156 "absence of charge inversions" does not support that the residues in "region 3" are less important than the "region 1" residues and should be removed.

4) Please describe the "gate" in SapA that allows access of galactocerebroside to the dimer cavity. W37 seems to be important here. Also, please describe the SapA/SapA contacts (with SI Figure?).

5) Please model the binding of monomeric "closed apo" SapA to GALC. The closed form seems to be incompatible with the complex, largely because of conformational changes in the hinge region affecting regions 1 and 3 (i.e. only head-to-head dimeric open SapA with a defined hinge angle can bind). This is interesting/relevant.

Minor comments

6) The MS should state that the four chains of the heterocomplex are related by a non-crystallographic two-fold symmetry axis. Were NCS restraints used during refinement? How similar are the two half-structures?

7) Please include the C α RMSDs with the individual MR models (PDB ID 4CCE and 4DDJ). A comment about the degree of hinge opening of the SapA chain would be helpful. It appears that the opening seen here is very similar to that in 4DDJ. This is remarkable, and suggests that the open form of SapA is quite rigid.

8) Fig. S2. I am puzzled about the fact that the LDAO difference density is negative. The result is quite clear, but I am not sure why the LDAO volume would have lower electron density than the bulk water phase. I suspect that the volume would be negative even if it were filled with bulk water instead of LDAO (but there is little doubt that it is filled with LDAO - I am just saying that this test does not prove it). Have the authors tried to do a few rounds of refinement without bulk solvent correction (possibly using a different low resolution cut-off) to see if the LDAO difference density is positive?

9) Figure S2(B) has little value, especially given the different head-to-tail arrangement in 4DDJ, and should be removed.

10) Page 3 line 41. Incorrect use of "polycistronic". Only one polypeptide (prosaposin) is made from the gene.

11) Fig. 1A. The pH range for this experiment is unfortunate (4.0-5.4, 7.4). A pH of 5.4/5.6 is then used for later pulldowns. It would have been better to show a coarser pH range that is centered at

~5.5 (4.0 to 7.5?).

12) The methods for the expression and purification of SapB and SapD (Fig. 4) are not described.

13) Fig 2A (and S4B) shows that the GALC-binding surface of SapA is hydrophobic, yet the region is highly charged and involves (mostly) polar residues (Fig. 3, 4, S3). This is confusing – please clarify.

14) Fig. 4. Panel A: Please include the murine SapA sequence in the alignment. Also indicate the positions of the α -helices. Panel C: Specify human or murine GALC/SapA/SapB/SapD. Are the GALC contact residues identical in human vs mouse?

15) Figs 3 and S3 indicate hydrogen bonds between W37/H237 and W37/Y238 in region 2. There are no hydrogens on any of the atoms involved, and so these are not hydrogen bonds. These are important polar/VdW contacts, however. Please check the geometries of all of the reported “hydrogen bonds” to be sure that they are in fact H-bonds.

Reviewer #3 (Remarks to the Author):

Hill et al review, June 20th, 2017

This manuscript describes the first co-crystal structure of a saposin (SapA) in complex with its hydrolytic enzyme partner (GALC). Overall, the study addresses an important topic, is well written, and is of general interest. Although the resolution is limited, the structure is well refined. Pull-down assays find the interaction between SapA and GALC is pH dependent, requires detergent, and is sensitive to salt concentration. The structure reveals a tetrameric complex with two SapA molecules sandwiched between two GALC molecules. Interestingly, the SapA dimer from the complex structure with GALC differs from the previously determined SapA/detergent structure alone. A plausible mechanism for substrate entry into the active site is proposed based on an open cavity from within the SapA dimer that leads to the GALC active site. The main conclusion is that this structure represents the biologically relevant SapA/GALC complex for galactocerebroside processing within lysosomes. Charge inversion point mutants of key contact residues in the SapA/GALC interface support the authors' hypothesis. The only major point to address is the enzymatic characterization of GALC and SapA on galactosylceramide hydrolysis.

Major point

1. To support the author's conclusion that this structure represents the biologically relevant complex, it is important to characterize the effects of purified SapA on purified GALC enzyme activity towards its endogenous substrate galactosylceramide. Does SapA stimulate GALC activity? If so, how is this affected by the SapA point mutants? If not, can the authors provide a rationale?

Minor points

1. Are there any pathogenic mutations in GALC or SapA that are found at the interface between SapA and GALC? And is there any new insight from this structure into lysosomal storage disease?

2. The text describes in detail the modeling of substrate into the open cavity, but this is hard to visualize in Fig. 2B. It would be useful to label the active site in GALC in Fig. 2B. In addition, it would be useful to show where the galactose headgroup of the substrate/product is physically (from previous structures) in relation to the open cavity/active site. Does the open cavity make sense with regards to

the chemical bond between galactose and ceramide that is hydrolyzed by GALC? Lastly, is the open cavity large enough to fit the substrate? Or are conformational changes required for galactosylceramide to bind within the cavity? The text appears to allude that minor changes are needed to accommodate galactosylceramide, if so, this point could be made more explicitly.

3. What does the crystal packing look like? And are there any crystal contacts that might affect the SapA/GALC interface?

4. In the discussion, there is a nice description of how the new SapA dimer compares to the previously determined SapA dimer. However, it would be helpful to cite Fig. S4A at the beginning of the description or alternatively include an additional supplemental figure explicitly comparing the different dimeric SapA structures.

5. A diagram of the enzyme reaction catalyzed by GALC would be useful for readers not familiar with this important process.

6. A schematic of the two differing functional models: solubilizer vs. lipase would also be useful for readers.

Suggestion

1. Although outside the scope of this work, does the endolysosomal lipid BMP affect SapA/GALC complex formation and the pH sensitivity of binding/hydrolysis?

Michael Airola, Ph.D.
Assistant Professor
Dept. of Biochemistry and Cell Biology
Stony Brook University

We are very pleased to note the comments from the reviewers that this represents a “landmark structure that explains a lot of preceding biology/biochemistry” and that the work “addresses an important topic, is well written, and is of general interest”. We thank the reviewers for their very helpful comments and have addressed them as detailed below.

Reviewer #1

Although this mechanism appears quite convincing based on the structure alone, it still needs to be tested, which is the major weakness of the paper - the model would be much stronger if it could be tested through mutagenesis of the proposed lipid binding cavity combined with, at the very least, an *in vitro* lipid hydrolysis assay. Is there any evidence *in vivo* (or even *in vitro*) that the two proteins form a heterotetrameric complex and is this complex catalytically active to begin with? Does SapA binding regulate the activity of GALC?

There is strong evidence *in vivo* supporting the importance of SapA for GALC activity. Clinical studies have identified a patient with Krabbe disease who did not have a mutation in GALC but instead had one in SapA (1). Specifically, deletion of V11 in SapA causes Krabbe disease despite normal GALC activity. In addition, a transgenic mouse with SapA lacking one of the three disulphide bridges develops a phenotype resembling the late-onset form of Krabbe disease (2). To date, *in vitro* studies of saposin-dependence do not recapitulate these known disease associations, i.e. GALC activity was shown to be enhanced by SapC but mutation of SapC causes Gaucher disease, not Krabbe disease (3-5). We agree with the reviewer that ideally a reliable *in vitro* assay for hydrolysis of lipidated substrates would be extremely useful for validating our structure and we have made extensive efforts to try and develop an assay that yields robust results. However, we have encountered a number of problems that we have not been able to overcome to our satisfaction. We are continuing to test and develop saposin-dependent *in vitro* and cell-based activity assays but such efforts are beyond the scope of this current study. We have updated the introduction to emphasise that there is clear evidence for the requirement of SapA to support GALC activity *in vivo*.

Regarding the reviewer’s question about whether there is evidence for a heterotetrameric complex *in vivo* or *in vitro*, we would argue that this structure provides the first *in vitro* evidence. We expect that this tetrameric complex exists only transiently as would be expected for what is essentially an enzyme-product complex and expand upon this in response to this reviewer’s point 2.

**Beyond this major point, the following are a list of minor points that should be addressed:
1) Can the authors explain based on the structure, why the closed form of SapA cannot interact with GALC?**

The closed form of SapA can be docked onto the open form of SapA via the terminal helices or the central two helices (Figure S7A, below). Only this second alignment yields an interface that maintains any of the critical interactions identified in our pulldowns (Figure S7B). This arrangement would block binding of a second SapA and GALC resulting in a 1:1 complex that possesses a buried surface area of 669 Å² (compared with 1366 Å² for the tetramer) and is not predicted by ePISA to form a stable complex (CSS=0, (6)). Thus, although there is no steric hindrance to binding the closed form it is likely to be an extremely weak interaction, below the detection limit of our assays if it exists at all. In response to this request and that of Reviewer 2 point 5, we have added a description of this modelling and the figure below as a supplementary figure.

Figure S7. Overlays of closed SapA with the open SapA in our structure. (A) Closed SapA (PDB ID 2DOB, (7)) was docked onto open SapA via superposition of the terminal helices (residues 1-20 and 65-80, left) or the central helices (residues 24-59). (B) Docking of closed SapA via the central helices of SapA within the context of the GALC-SapA complex would block formation of a heterotetramer, but would maintain some of the critical interactions identified in this study. (C) The docking of closed SapA onto a monomer of GALC buries a surface area of 669 Å² (compared with 1366 Å² for the tetramer) and is not predicted by ePISA to form a stable complex (CSS=0, (6)).

2) There is no discussion on what could occur after lipid hydrolysis is complete. What regulates docking and undocking of SapA from GALC?

Although we were able to identify conditions where the GALC-SapA interaction could be monitored using pulldown experiments, the interaction is likely to be relatively weak as the wash steps had to be kept short and the complex dissociated during size-exclusion chromatography. These observations are what we might expect for a complex that should be transient such that following substrate processing the enzyme and saposin complex dissociate. To monitor these weak interactions biophysically proved very challenging despite attempts to enhance complex formation using either non-hydrolysable acylated inhibitors or the catalytically defective E258Q variant with substrates. Thus, our hypothesis is that the interaction is weak and undocking of the SapA from GALC would occur readily without an intact substrate to bridge the interface. Ongoing work in the lab is testing this hypothesis but without reliable measurements of the affinity we prefer not to speculate regarding this in the discussion.

3) Does lipid access to the active site occur symmetrically on both protomers? In other words, does the SapA dimer have enough room for two lipid molecules? It is not clear from Supp. Fig. S2 if this is the case. Or can only one catalytic event occur at a time?

Lipid access to the active site could occur symmetrically. Based on the ordered LDAO molecules modelled in the lipoprotein disc structure (PDB ID 4DDJ, (8)) we can position between 10 and 14 LDAO molecules in the cavity of our dimer (Figure R1). Each LDAO contains 12 carbon atoms in its hydrophobic chain, suggesting the cavity in our structure could accommodate approximately 120 to 168 atoms as saturated hydrocarbon chains. The length of the acyl chains in the ceramide portion of GalCer can vary but the example presented in the new Figure 1 (see response to Reviewer 3 point 6) illustrates an 18-carbon fatty acid tail and a 15-carbon sphingosine chain totalling 33 carbon atoms for a single GalCer molecule. Thus, the cavity could accommodate at least 2 GalCer molecules while also allowing for variations in chain length and conformation. A simplified version of the figure below illustrating how many LDAO molecules can fit in the cavity has been added as a panel of Figure S3 and a description of this cavity volume and the capacity to bind two GalCer molecules has been added to the main text.

Figure R1. The volume of the SapA dimer cavity can accommodate 10-14 LDAO molecules. (A) Ribbon diagram showing two orientations of the SapA dimer with 14 molecules of LDAO positioned based on the lipoprotein disc structure (PDB 4DDJ). (B) As for panel A but illustrating the surface of SapA. (C and D) Illustration of the equivalent orientations and surfaces of the original lipoprotein disc structure.

4) The R -factors (particularly R_{free}) are unusually low for 3.6 Å resolution. I realize the refinement was restrained to the high resolution structure of GALC, but was this also done for SapA? How was the R_{free} test set chosen? Also in Table 1: the Wilson B -factor should be added to the data collection statistics.

SapA was not refined with restraints to the high resolution structure, this was only done for the GALC chains. The reviewer is correct that the low residuals are due to this restrained refinement to the high resolution GALC structure and as GALC contributes 89% of the scattering mass, it contributes the bulk of the phase information. In addition, NCS restraints were included between SapA and GALC molecules as detailed in response to Reviewer 2 point 6. The R_{free} set was selected as a random set. Given that this crystal is a different lattice from the high resolution structures, these reflections will be independent. The Wilson B is now included in Table 1.

5) Title: "The mechanism of sphingolipid processing revealed by a GALC-SapA complex structure" - is a bit overreaching. Various sphingomyelinases and ceramidases employ different mechanisms. The authors should consider something more specific such as: "The mechanism of glycosphingolipid degradation..."

This is a very fair point and we have altered the title as suggested.

6) Introduction: "lipid-transfer proteins including saposins and GM2 activator proteins" - there is only one GM2 activator protein.

We have amended the text appropriately.

7) Introduction: "polycistronic prosaposin" - polycistronic refers to multiple open reading frames on one mRNA. It should be changed to "precursor prosaposin".

We have amended the text appropriately.

8) Results: "In addition to electrostatic interactions, hydrophobic interactions at the interface are also critical for binding. The sidechain of L28 interacts with a hydrophobic surface of GALC (Figure S3C) and mutation of this residue to arginine abolishes SapA binding to GALC." This mutation causes

likely results in unavoidable steric clashes with GALC, so the importance of the 'hydrophobic' nature of the interaction involving L28 cannot be assessed by it.

We have now made an L28N mutation that maintains the sidechain length but not the hydrophobicity. In pulldown experiments with GALC this more subtle mutation also significantly reduces binding supporting the importance of the hydrophobic nature of this interaction. We have added this new data as an additional gel in Fig 4B (reproduced below in response to this reviewer point 14).

9) Results: "Residue N21 lies at the interaction interface and forms a backbone hydrogen bond with G144 and a number of hydrophobic interactions with the GALC surface. However, in the context of the cell this highly-conserved residue is post-translationally glycosylated. N-linked glycans often interact with hydrophobic patches on the surface of proteins (29) and mutation of N21 to the bulky hydrophobic sidechain tyrosine moderately increased SapA binding to GALC, suggesting that the glycosylation of N21 contributes to SapA-GALC binding in vivo." N21 is facing GALC with its polar side chain, and that area of GALC is mainly lined with backbone atoms. Water molecules are present there, in the high resolution structure of GALC (PDB 4CCD). Therefore, hydrophobic interactions are probably not at play here; is there evidence that glycosylation of N21 actually contribute to complex formation?

While it is true that the area of GALC with which N21 interacts is lined with backbone atoms, the peptide bonds between GALC residues 144-146 are coplanar and there is thus a significant hydrophobic surface against which the tyrosine ring (and the atoms of the sugar ring) could pack. The position of residue N21 was of interest to us as it binds the lectin domain near the glycan-binding site of homologous lectin domains, suggesting the glycan on this residue may contribute to the interaction. However, as there is no other supporting data for a contribution of the glycosylation to the binding we have re-written this section in order not to overstate the potential role of this modification.

10) Discussion: "The interaction of SapA with GALC is pH dependent" - based on the structure of the complex, is there an explanation for this?

The overall charge of GALC changes from $-5 e^-$ to $+15 e^-$ between pH 7.5 and pH 5.4, and specifically the surface that is involved in the interaction with SapA becomes highly positively charged (Figure S2 below). The low pI of SapA means that it retains an overall negative charge at both pH values, $-16 e^-$ to $-10 e^-$. Thus, the transition in overall surface charge for GALC from 7.5 to 5.4 is likely to play a part in the pH dependence of the interaction. We have included the figure below as an additional supplementary figure and added a description to the main text.

Figure S2. Electrostatic surface potential of GALC and SapA at (A) pH 7.5 and (B) pH 5.4. The overall charge of GALC changes from $-5 e^-$ to $+15 e^-$ between pH 7.5 and pH 5.4, while SapA retains an overall negative charge at both pHs, $-16 e^-$ to $-10 e^-$. This change in surface charge for GALC is particularly prevalent across the interaction interface with SapA. Thus, the transition in overall surface charge for GALC from 7.5 to 5.4 is likely to play a part in the pH dependence of this interaction.

11) Discussion: "Specifically, SapA is thought to function as a 'solubiliser', an assertion strongly supported by the GALC- SapA structure presented here." - References should be added that support this model (such as #21, where soluble SapA lipoprotein particles are described).

We have amended the text appropriately.

12) Figure 2: the channel connecting the GALC active site to the hydrophobic cavity in the saposin dimer seems to be narrowed down by the side chain of saposin Y54. Is this the case? Does the substrate still fit? Could rotation of this side chain facilitate substrate entry?

The dimensions of the channel at the narrowest point, taking into account van der Waals radii, would be a tight fit for two acyl chains. We agree with the reviewer that Y54 is playing a role in shaping the channel and specifically may be narrowing the passage in its observed conformation. Our previous GALC structure with non-lipid substrate has shown that the enzyme has significant conformational flexibility at the active site. We think that movements around the channel, including rotation of Y54 and other sidechains, may help facilitate substrate entry. As part of our response to this point and to reviewer 2 point 4 we have added a more extensive description of this opening and included a new panel in figure 3 illustrating the residues that are critical in forming this opening.

13) Figure 3A: the legend should specify that some SapA residues interact with multiple GALC residues; otherwise parts of the diagram could be interpreted as interactions within GALC residues.

The legend has been amended appropriately.

14) Fig. 3B: has W37 also been mutated, to assess the importance of the secondary interaction surface?

In response to this query and to Reviewer 2 points 1 and 2 (below) we have made three different mutations of W37 to test the importance of this residue at the interface. The mutations of W37 were designed to test the importance of the hydrophobic interactions (W37D), the hydrogen bonding (W37F) and the combination of these interactions (W37S). All three mutations were tested three times independently and maintained binding to GALC in our pulldown experiments. Thus, within the context of our pulldown experiments, this secondary interaction surface is less important. However, as W37 forms part of the opening through which the substrate must pass it is likely this residue is still critical for function if not essential for binding. To test this we need to develop a reliable saposin-dependent activity assay as described in response to reviewer 1 point 1, which is beyond the scope of this current study. The new pulldown data for W37 has been added to the main paper as a new gel in Fig. 4B (reproduced below) and the discussion of this interface re-written to incorporate this new data and address concerns of reviewer 2 below.

Figure 4B (new panel). Coomassie-stained SDS-PAGE of pulldown assays with immobilised GALC and new mutations of SapA: L28N, W37D, W37S and W37F.

Reviewer #2

Major comments:

1) Several SapA residues are tested for the importance with their pulldown assay (Fig. 3). The results are convincing (unfortunately, it would be more difficult to carry out a similar test of the GALC surface-exposed residues since GALC is produced in mammalian cells - SapA is expressed in *E. coli*). The GALC-SapA interface has many electrostatic salt bridges between complementary residues and this is extensively documented and discussed. However, the interface is more complex than charge complementarity and contacts that appear to be important are glossed over. In particular, the “region 2” surface of SapA (SapA-2/GALC) appears to be highly important but is largely dismissed because it involved mostly backbone residues. This is an oversimplification – this surface of SapA has excellent shape complementarity to the GALC surface and contributes a large percentage of the buried surface. I disagree with the statement: “... residues 36-40 primarily participates in backbone interactions so does not determine specificity” (lines 152-153). The side chains in this region have important roles in positing the backbone residues for complex formation. Overall, I would suggest a more balanced interpretation of all three SapA regions of the binding surface (see also points 2 and 3 below).

The description of the interface both in the results and discussion has been re-written to address the concerns of reviewer 2.

2) Of particular note, the side chain of SapA W37 in region 2 makes extensive contacts with GALC (Figs 3, S3). Can this residue be tested with a mutant? The mutants tested in Fig. 3 cover one chain only.

As detailed above in response to reviewer 1 point 14, we have made three new mutations: W37S, W37D and W37F. All three maintain binding to GALC in the context of our pulldown assays. This data and a description of these new mutations has been added to the main text.

3) “Charge inversion” between the saposins is overemphasized throughout - this implies negative selection of the other saposins to GALC. Given the low sequence identity between the four saposins, it is just as likely that the differences are purely due to neutral drift or positive selection to the partner enzymes. The argument in lines 153-156 “absence of charge inversions” does not support that the residues in “region 3” are less important than the “region 1” residues and should be removed.

This text has been removed and re-written as part of our response to point 1 above.

4) Please describe the “gate” in SapA that allows access of galactocerebroside to the dimer cavity. W37 seems to be important here.

As a response to this point and to reviewer 1 point 12 above we have added a description of the opening in the SapA dimer and included an additional panel to Figure 3. For additional clarity here we have included this panel below (right) and an illustration of this opening within the context of the dimer. We agree that W37 is likely to play an important role here and have included this statement in our description of the new W37 mutations as described in response to Reviewer 1 point 14.

Figure R2. Illustration of the opening in the SapA dimer. Residues that form the opening at the SapA dimer surface opposite the GALC active site are shown as surface on the ribbon diagram of the dimer (left panel). Zoomed in view of this opening with residues shown as sticks with a semi-transparent surface.

4) continued...Also, please describe the SapA/SapA contacts (with SI Figure?).

Analysis of the SapA-SapA interface using ePISA identifies that the protein contacts are primarily hydrophobic with the most energy contributed by burying the sidechains of hydrophobic residues clustered at one end of the dimer, specifically L2, I6, V10, L76 and L78. Additional contributions to the interface are made at the limiting edges of the opening opposite the GALC active site and involve residues Y30, I38, Y54, M61 and V50. A figure illustrating these interactions has been added as one panel of a new Figure S5 that compares the SapA dimer structures as requested by reviewer 3 point 4 (where the new figure is reproduced in full). The main text describing this dimer has also been expanded.

5) Please model the binding of monomeric “closed apo” SapA to GALC. The closed form seems to be incompatible with the complex, largely because of conformational changes in the hinge region affecting regions 1 and 3 (i.e. only head-to-head dimeric open SapA with a defined hinge angle can bind). This is interesting/relevant.

In response to this request and to that of Reviewer 1 point 1 we have included a description and supplementary figure describing docking of monomeric “closed apo” SapA onto GALC. In summary here, the closed form can be docked onto GALC forming a 1:1 complex that maintains some crucial interactions. However, the buried surface area is much smaller and not predicted to be significant suggesting this complex, if it exists, is extremely weak.

A discussion of the hinge angle is detailed below in response to this reviewer’s point 7.

Minor comments

6) The MS should state that the four chains of the heterocomplex are related by a non-crystallographic two-fold symmetry axis. Were NCS restraints used during refinement? How similar are the two half-structures?

NCS restraints were used during refinement and the methods sections has been updated to clarify this point. We have also added a line to the main text clarifying that the heterotetramer is formed by a pseudo two-fold symmetry axis. The two half-structures are very similar with RMSDs of 0.16 Å between GALC chains (calculated over 644 C α) and 0.19 Å between SapA chains (calculated over 80 C α). Overlay of one half of the structure with the other half reveals a small (3.7°) rotation about the pseudo two-fold axis (Figure R3).

Figure R3. Superposition illustrating the non-crystallographic symmetry. The structure of the tetramer was overlaid on itself to illustrate the small 3.7° rotation about the pseudo two-fold axis.

7) Please include the Calpha RMSDs with the individual MR models (PDB ID 4CCE and 4DDJ). A comment about the degree of hinge opening of the SapA chain would be helpful. It appears that the opening seen here is very similar to that in 4DDJ. This is remarkable, and suggests that the open form of SapA is quite rigid.

We agree with the reviewer that it is intriguing and potentially important that the SapA monomer adopts the same open conformation and identical hinge angle upon formation of this different dimer. We have included the RMSD of the SapA monomers and also added a specific point about this similarity to the text as well as added a relevant panel to the new Figure S5 as part of our response to this reviewer point 4 and to reviewer 3 point 4 (where this full figure including monomer overlays is reproduced).

The RMSD between the GALC molecules and the high-resolution GALC structure (4CCE) is 0.29-0.32 Å over 640 C α atoms. However, we haven't included this measure in the main text as the high-resolution structure was used as reference model during refinement and thus the RMSD value is not independently determined.

8) Fig. S2. I am puzzled about the fact that the LDAO difference density is negative. The result is quite clear, but I am not sure why the LDAO volume would have lower electron density than the bulk water phase. I suspect that the volume would be negative even if it were filled with bulk water instead of LDAO (but there is little doubt that it is filled with LDAO – I am just saying that this test does not prove it). Have the authors tried to do a few rounds of refinement without bulk solvent correction (possibly using a different low resolution cut-off) to see if the LDAO difference density is positive?

The figure shows the density with bulk solvent modelled in the interior, this is why we specify that we used the "AnalyseVoids=no" option to ensure that Buster did not exclude this area from the bulk solvent model. The observation of negative density in the cavity is consistent with the expectation that the disordered acyl chains would have a lower average electron density than the bulk solvent, as has been observed in a previous crystal structure that included significant regions of disordered lipid (9). We have not calculated a map following refinement without a bulk solvent model using, for example, a low resolution cutoff of 6Å (i.e. including only 6–3.6Å data). This is because we do not have evidence that any of the LDAO acyl chains are ordered, and such acyl chains would thus not be expected to contribute significantly to the intensities measured for high resolution reflections. By excluding low resolution reflections, we would be excluding the reflections that carry the most information about the regions of the asymmetric unit that are not occupied by ordered protein molecules (10).

9) Figure S2(B) has little value, especially given the different head-to-tail arrangement in 4DDJ, and should be removed.

This panel and the reference to it in the main text have been removed.

10) Page 3 line 41. Incorrect use of "polycistronic". Only one polypeptide (prosaposin) is made from the gene.

Corrected as also requested by reviewer 1.

11) Fig. 1A. The pH range for this experiment is unfortunate (4.0-5.4, 7.4). A pH of 5.4/5.6 is then used for later pulldowns. It would have been better to show a coarser pH range that is centered at ~5.5 (4.0 to 7.5?).

We have redone the pulldown across a broader pH range from pH 4.0 to pH 7.5. We have repeated this pulldown three times and quantified the binding across this pH range (Figure R4, below). The interaction is optimal at pH 5.5, the band intensity being 3.6-fold greater at pH 5.5 compared with pH 7.5. Below pH 4.5 there was considerable elution of GALC from the nickel resin and so the loss of SapA binding here is due to loss of the bait protein. This new pulldown data with a broader pH range has replaced the previous pH binding panel of Fig. 2.

Figure R4. Quantification of SapA binding to GALC across a pH range from 7.5 to 4.0. (A) Coomassie-stained SDS-PAGE of pulldown assay with immobilised GALC across the pH range 7.5 to pH 4.5 in the presence of the detergent. (B) Quantification of SapA binding in pulldown assays (n=3, error bars represent SEM). The intensity of SapA shows binding to GALC is 3.6 times greater at pH 5.5, values equivalent to the late-endosomal/lysosomal compartment.

12) The methods for the expression and purification of SapB and SapD (Fig. 4) are not described.

mSapB and mSapD were purified using identical protocols to that used for SapA. The methods have been updated to include this detail.

13) Fig 2A (and S4B) shows that the GALC-binding surface of SapA is hydrophobic, yet the region is highly charged and involves (mostly) polar residues (Fig. 3, 4, S3). This is confusing – please clarify.

We apologise that these figures were confusing and have adjusted the description of these to help clarify the points being made. There were two potential confusions that we have addressed. The first is that the right-most panel of figure 3A (formerly Fig 2A) shows the very hydrophobic surface of SapA that is buried in the SapA dimer, not the surface that interacts with GALC. The second point is that the central panel shows the surface of SapA that is involved in the GALC interaction, illustrating a mixture of polar and hydrophobic interactions, with a small portion circled just highlighting the region that lies over the active site. We have altered the text referencing this figure to clarify these points.

14) Fig. 4. Panel A: Please include the murine SapA sequence in the alignment. Also indicate the positions of the α -helices. Panel C: Specify human or murine GALC/SapA/SapB/SapD. Are the GALC contact residues identical in human vs mouse?

Figure 5 (formerly Figure 4) has been modified to include both human and mouse saposin sequences in the alignment. The position of α -helices have been added and panel C updated with species (all murine). The GALC contact residues are identical except for residue S146, which is an Asp in human GALC, and Y382, which is a Phe. This first difference (S146D) would not result in any steric hindrance and may actually make the complex with human GALC tighter as an Asp at this position may be able to form a salt bridge with SapA K63 while the mouse GALC can only make hydrogen bonds. The second change (Y382F) is highly conservative and this residue only participates in backbone hydrogen bonds with SapA so is unlikely to have any effect.

15) Figs 3 and S3 indicate hydrogen bonds between W37/H237 and W37/Y238 in region 2. There are no hydrogens on any of the atoms involved, and so these are not hydrogen bonds. These are important polar/VdW contacts, however. Please check the geometries of all of the reported “hydrogen bonds” to be sure that they are in fact H-bonds.

The indole nitrogen of tryptophan is protonated and can engage in hydrogen bonds with the carbonyl oxygen atoms of H237 and Y238. The geometry at this site is consistent with hydrogen bond formation as these atoms are co-planar with the indole ring of W37, and thus the proton attached to the indole nitrogen.

The geometries of all hydrogen bonds have been checked and figures updated to ensure that all interactions reported in Figures 4 (formerly Fig 3) and S4 (formerly Fig S3) are correct.

Reviewer #3

1. To support the author’s conclusion that this structure represents the biologically relevant complex, it is important to characterize the effects of purified SapA on purified GALC enzyme activity towards its endogenous substrate galactosylceramide. Does SapA stimulate GALC activity? If so, how is this affected by the SapA point mutants? If not, can the authors provide a rationale?

We refer the reviewer to our response to Reviewer 1 point 1.

Minor points

1. Are there any pathogenic mutations in GALC or SapA that are found at the interface between SapA and GALC? And is there any new insight from this structure into lysosomal storage disease?

There are two disease-causing mutations that we have previously identified as being correctly folded (i.e. are trafficked beyond the ER and are therefore not misfolding mutations) and retain activity in assays using water-soluble substrates (11, 12). These two mutations lie very near the interface with SapA and so the pathologic mechanism of these variants may be inability to bind SapA (Figure S8, reproduced below). In the original manuscript we did not describe these as they do not make direct interactions with SapA. However, considering their proximity to the interface it seems likely that they alter the surface shape and surface charge such that the interaction could be reduced or abolished. Due to the potential interest of this insight we have now added this figure as an additional supplementary item and added a brief discussion to the main text. Future work will directly test this by producing these variants and carrying out pulldowns and, ideally, activity assays. However, as Reviewer 2 rightly points out it will take some time to produce these proteins as they are expressed in mammalian cells and we will need to generate new stable, high-expressing cell lines.

Figure S8. Two disease-causing mutations of GALC lie near the interaction interface with SapA. (A) Ribbon diagram of the GALC-SapA complex illustrating the position of two disease-causing mutations E215K and P302R (red sticks). (B) Surface diagram illustrating the interaction surface of GALC as shown in Figure 2D highlighting the proximity of the two disease-causing mutations E215K and P302R (red surface with dashed outline) to the saposin-binding epitope.

2. The text describes in detail the modeling of substrate into the open cavity, but this is hard to visualize in Fig. 2B. It would be useful to label the active site in GALC in Fig. 2B. In addition, it would be useful to show where the galactose headgroup of the substrate/product is physically (from previous structures) in relation to the open cavity/active site. Does the open cavity make sense with regards to the chemical bond between galactose and ceramide that is hydrolyzed by GALC? Lastly, is the open cavity large enough to fit the substrate? Or are conformational changes required for galactosylceramide to bind within the cavity? The text appears to allude that minor changes are needed to accommodate galactosylceramide, if so, this point could be made more explicitly.

A new panel has been added to figure 3 that illustrates more clearly how and where the galactose portion of the substrate binds based on the previously determined structure of GALC with bound water-soluble substrate (PDB 4CCC). The position of the galactose and the scissile bond are fully compatible with hydrolysis and would position the acyl chains appropriately to enter the hydrophobic channel. As discussed in response to reviewer 1 points 12 and 3 the cavity within the SapA dimer is easily large enough to accommodate substrate and the opening in the dimer surface (also an additional panel in Fig 3) is, in its current conformation, a tight fit for 2 acyl chains. The text describing this has been updated to more explicitly describe the residues that may be involved in conformational changes to more easily accommodate lipidated substrate.

3. What does the crystal packing look like? And are there any crystal contacts that might affect the SapA/GALC interface?

The figure below illustrates the crystal packing arrangement and shows that there are no contacts affecting the GALC-SapA interface. This has been added as an additional panel in Figure S1 and referenced in the main text.

Figure S1E. COOT screenshot illustrating the symmetry related molecules in the crystal lattice. For clarity, all chains are shown as backbone traces.

4. In the discussion, there is nice description of how the new SapA dimer compares to the previously determined SapA dimer. However, it would be helpful to cite Fig. S4A at the beginning of the description or alternatively include an additional supplemental figure explicitly comparing the different dimeric SapA structures.

As a response to this request and that of Reviewer 2 points 4 and 7 we have now made an additional Supplementary Figure, reproduced below, clearly comparing the two SapA dimers.

Figure S5. Comparison of SapA dimer structures. (A) The conformation of the SapA monomers in the GALC complex (yellow) and the lipoprotein disc (cyan, (8)) are almost identical, except for a small loop movement at residues W37 and I38 (*). (B) In the dimer structures the second SapA chains (orange and pink, respectively) are in different positions and opposite orientations (N and C termini marked with circles) with the lipoprotein disc structure possessing a larger cavity. (C) The SapA dimer in complex with GALC possesses several protein-protein contacts, including multiple hydrophobic interactions where the termini come together (L2, I6, V10, L76 and L78) and contacts near the interface with GALC (Y30, I38, V50, Y54 and M61). (D) The lipoprotein disc does not possess direct protein-protein interactions and is instead maintained via interactions with ordered LDAO molecules (yellow sticks).

5. A diagram of the enzyme reaction catalyzed by GALC would be useful for readers not familiar with this important process AND

6. A schematic of the two differing functional models: solubilizer vs. liftase would also be useful for readers.

We have now added an additional Introductory figure (Fig. 1, reproduced below) that includes both the reaction catalysed and the different proposed models for saposin function as requested in the two points above.

Figure 1. Glycosphingolipid processing by saposins and GALC. (A) Schematic diagram for the two proposed mechanisms of saposin-mediated hydrolase activation. The solubiliser model (left) proposes saposins (yellow) extract sphingolipids from the bilayer forming a soluble saposin-lipid complex that presents the lipid to the hydrolase (blue). The liftase model (right) proposes saposins (orange) disrupt or insert into the bilayer to provide access for hydrolases (purple) to the lipid substrates at the bilayer surface. Glycosyl headgroups are illustrated as green hexagons. (B) The cleavage of galactocerebroside (GalCer) by GALC produces galactose and ceramide.

Suggestion

1. Although outside the scope of this work, does the endolysosomal lipid BMP affect Sapa/GALC complex formation and the pH sensitivity of binding/hydrolysis?

This is an intriguing suggestion and something we will endeavour to address in future work.

References

1. Spiegel R, Bach G, Sury V, Mengistu G, Meidan B, Shalev S, Shneur Y, Mandel H, & Zeigler M (2005) A mutation in the saposin A coding region of the prosaposin gene in an infant presenting as Krabbe disease: first report of saposin A deficiency in humans. *Mol Genet Metab* 84:160-166.
2. Matsuda J, Vanier MT, Saito Y, Tohyama J, & Suzuki K (2001) A mutation in the saposin A domain of the sphingolipid activator protein (prosaposin) gene results in a late-onset, chronic form of globoid cell leukodystrophy in the mouse. *Hum Mol Genet* 10:1191-1199.
3. Vaccaro AM, Tatti M, Ciaffoni F, Salvioli R, Barca A, & Scerch C (1997) Effect of saposins A and C on the enzymatic hydrolysis of liposomal glucosylceramide. *J Biol Chem* 272:16862-16867.
4. Harzer K, Paton BC, Christomanou H, Chatelut M, Levade T, Hiraiwa M, & O'Brien JS (1997) Saposins (sap) A and C activate the degradation of galactosylceramide in living cells. *FEBS Lett* 417:270-274.
5. Harzer K, Hiraiwa M, & Paton BC (2001) Saposins (sap) A and C activate the degradation of galactosylsphingosine. *FEBS Lett* 508:107-110.
6. Krissinel E & Henrick K (2007) Inference of macromolecular assemblies from crystalline state. *J Mol Biol* 372:774-797.
7. Ahn VE, Leyko P, Alattia JR, Chen L, & Prive GG (2006) Crystal structures of saposins A and C. *Protein Sci* 15:1849-1857.
8. Popovic K, Holyoake J, Pomes R, & Prive GG (2012) Structure of saposin A lipoprotein discs. *Proc Natl Acad Sci U S A* 109:2908-2912.
9. Cockburn JJ, Abrescia NG, Grimes JM, Sutton GC, Diprose JM, Benevides JM, Thomas GJ, Jr., Bamford JK, Bamford DH, & Stuart DI (2004) Membrane structure and interactions with protein and DNA in bacteriophage PRD1. *Nature* 432:122-125.
10. Bragg WL & Perutz MF (1952) The external form of the haemoglobin molecule. I. *Acta Crystallogr.* 5:277-283.

11. Spratley SJ, Hill CH, Viuff AH, Edgar JR, Skjodt K, & Deane JE (2016) Molecular Mechanisms of Disease Pathogenesis Differ in Krabbe Disease Variants. *Traffic* 17:908-922.
12. Hill CH, Graham SC, Read RJ, & Deane JE (2013) Structural snapshots illustrate the catalytic cycle of beta-galactocerebrosidase, the defective enzyme in Krabbe disease. *Proc Natl Acad Sci U S A* 110:20479-20484.

Reviewers' comments:

Reviewer #1 (Remarks to the Author):

I have looked through the author responses. It is understandable that the authors could not develop a robust in vitro activity assay with lipid substrates. However, I still would have liked to see activity measurements perhaps with a nitrophenyl-galactoside analogue, which although will not test their complex structure model, can at least show that their purified GalC is functional in the presence of saposin. All other responses to my queries are adequate.

Reviewer #2 (Remarks to the Author):

The authors have made extensive changes to their MS, and the revised version is considerably improved. My one remaining issue is the comparison of the SapA dimers of the new structure with a previous "lipoprotein disc structure" (ref 21). The lipoprotein discs from Ref 21 are now referred to as "picodiscs" (see, for example, PMID: 27532319).

While the conformations of the open monomers are nearly identical, there are major differences in how these are arranged in the dimers. This is interesting, but the picodisc quaternary structure is determined exclusively by LDAO-LDAO interactions, and is thus not physiological. In fact, ref 21 makes this point and includes MD results that show that the picodisc structure is highly plastic and that both head-to-head and head-to-tail configurations are possible. Plasticity was also described in PMID: 27532319. Please revise the text on pages 11-12 accordingly. Also, I do not think that Supplementary Figure S6 is relevant and could be deleted.

Reviewer #3 (Remarks to the Author):

The additions satisfy all the minor points. Thank you. However, there is still some concern with regards to the ability of SapA to stimulate GALC activity. Based on the references provided, it appears both SapA and SapC stimulate GALC activity, and potentially work synergistically through different mechanisms. Was SapC tested for binding to mGALC in Figure 5? In the author's opinion or based on their work, does SapA work synergistically or independently from SapC? Can SapA alone activate GALC? It appears to be an important issue to resolve and/or discuss to highlight the importance and role of the SapA-GALC complex structure to glycosphingolipid processing, as well as to address the specificity mentioned for SapA to GALC.

Response to Reviewers

Reviewer #1 (Remarks to the Author):

I have looked through the author responses. It is understandable that the authors could not develop a robust in vitro activity assay with lipid substrates. However, I still would have liked to see activity measurements perhaps with a nitrophenyl-galactoside analogue, which although will not test their complex structure model, can at least show that their purified GalC is functional in the presence of saposin. All other responses to my queries are adequate.

We have carried out activity assays in the presence of SapA and the GALC is still active. However, its activity is altered (in the presence of 100-fold excess of SapA, GALC activity drops by 50%). We hypothesised it might be a competitive inhibitor (as it binds across the active site) but analysis of the data isn't that simple. In addition to a drop in V_{max} , the shape of the V_o vs [substrate] curve doesn't fully support a simple competitive inhibitor model. It may be that SapA sequesters detergent and/or substrate but, in agreement with our previous response, such assays with saposins are not robust and are not appropriate for publication at this stage. As the reviewer states, these assays do not test the presented complex structure and so further work to optimise these assays is outside the scope of the current manuscript.

Reviewer #2 (Remarks to the Author):

The authors have made extensive changes to their MS, and the revised version is considerably improved. My one remaining issue is the comparison of the SapA dimers of the new structure with a previous "lipoprotein disc structure" (ref 21). The lipoprotein discs from Ref 21 are now referred to as "picodiscs" (see, for example, PMID: 27532319).

At first reference to the lipoprotein disc structure we will include that these are also referred to as picodiscs and also as Salipro nanoparticles (as this is the term used in the Nature Methods paper referenced in the Introduction). However, we want to retain the use of the term lipoprotein disc in the text as this will make most sense to readers wanting to understand the comparison with the structure published in ref 21 (now ref 22).

While the conformations of the open monomers are nearly identical, there are major differences in how these are arranged in the dimers. This is interesting, but the picodisc quaternary structure is determined exclusively by LDAO-LDAO interactions, and is thus not physiological. In fact, ref 21 makes this point and includes MD results that show that the picodisc structure is highly plastic and that both head-to-head and head-to-tail configurations are possible. Plasticity was also described in PMID: 27532319. Please revise the text on pages 11-12 accordingly. Also, I do not think that Supplementary Figure S6 is relevant and could be deleted.

We have rewritten this part of the manuscript to make it clear that the previous work has identified this plasticity in the LDAO-mediated discs. However, we have retained Figure S6 (now Figure S7) as in the previous round of reviewing Reviewer 3 point 4 explicitly stated that this figure (at that stage called Figure S4) was useful for understanding the comparison with the previously published SapA disc structure.

Reviewer #3 (Remarks to the Author):

The additions satisfy all the minor points. Thank you. However, there is still some concern with regards to the ability of SapA to stimulate GALC activity. Based on the references provided, it appears both SapA and SapC stimulate GALC activity, and potentially work synergistically through different mechanisms. Was SapC tested for binding to mGALC in Figure 5?

As stated in the manuscript we are unable to purify mouse SapC. However, we have recently been able to produce small amounts of human SapC and human GALC. Using these reagents we have carried out pulldowns which show that, unlike SapA, SapC does not bind GALC in this assay (Figure S5). We have included description of this data in the results and updated the methods (also see text below).

Figure S5. GALC binding to SapA but not SapC. (A) Coomassie-stained SDS-PAGE pulldown with immobilised murine GALC (mGALC) illustrating specific binding to murine SapA (mSapA), but not human SapC (hSapC). (B) Coomassie-stained SDS-PAGE pulldown with immobilised human GALC (hGALC) illustrating binding to both human and murine SapA, but not to hSapC.

In the author's opinion or based on their work, does SapA work synergistically or independently from SapC? Can SapA alone activate GALC? It appears to be an important issue to resolve and/or discuss to highlight the importance and role of the SapA-GALC complex structure to glycosphingolipid processing, as well as to address the specificity mentioned for SapA to GALC.

Our pulldown data (above) suggests that SapC does not directly bind GALC. Therefore, if SapC stimulates GALC activity it must do so via a different mechanism. SapC may function to enhance lipid accessibility at the bilayer surface, perhaps by helping load SapA. However, our data do not provide direct evidence to support this. Furthermore, as stated in our previous response to reviewer 1 regarding saposin dependency, clinical data strongly support correct SapA function in the absence of SapC as mutations in SapC result in Gaucher disease, not Krabbe disease. Indeed, we have now also included an additional reference identifying another SapC mutation that specifically causes Gaucher disease (19). To address this reviewer's questions we have written an additional discussion regarding this clinical data, the in vitro work and speculating upon possible roles of SapC.